# Comparative Transcriptomic and Metabolomic Analyses of Differences in Trunk Spiral Grain in *Pinus yunnanensis*

**DOI:** 10.3390/ijms241914658

**Published:** 2023-09-28

**Authors:** Peihua Gan, Peiling Li, Xiaolin Zhang, Hailin Li, Shaojie Ma, Dan Zong, Chengzhong He

**Affiliations:** 1Key Laboratory for Forest Genetics and Tree Improvement and Propagation in Universities of Yunnan Province, Southwest Forestry University, Kunming 650224, China; ganpeihua33@163.com (P.G.); ynlipeiling@163.com (P.L.); sunnyxlz27@163.com (X.Z.); hailinli1011@163.com (H.L.); mashaojie1013@163.com (S.M.); 2Key Laboratory for Forest Resources Conservation and Utilization in the Southwest Mountains of China, Ministry of Education, Southwest Forestry University, Kunming 650224, China; 3Key Laboratory of Biodiversity Conservation in Southwest China, State Forestry Administration, Southwest Forestry University, Kunming 650224, China

**Keywords:** spiral grain, twist, differential metabolites, gene expression, plant hormone signal transduction, terpenoid, *Pinus yunnanensis*

## Abstract

Having a spiral grain is considered to be one of the most important wood properties influencing wood quality. Here, transcriptome profiles and metabolome data were analyzed in the straight grain and twist grain of *Pinus yunnanensis*. A total of 6644 differential expression genes were found between the straight type and the twist type. A total of 126 differentially accumulated metabolites were detected. There were 24 common differential pathways identified from the transcriptome and metabolome, and these pathways were mainly annotated in ABC transporters, arginine and proline metabolism, flavonoid biosynthesis, isoquinoline alkaloid biosynthesis, linoleic acid metabolism, phenylpropanoid, tryptophan metabolism, etc. A weighted gene coexpression network analysis showed that the lightblue4 module was significantly correlated with 2′-deoxyuridine and that transcription factors (basic leucine zipper (bZIP), homeodomain leucine zipper (HD-ZIP), basic helix–loop–helix (bHLH), p-coumarate 3-hydroxylase (C3H), and N-acetylcysteine (NAC)) play important roles in regulating 2′-deoxyuridine, which may be involved in the formation of spiral grains. Meanwhile, the signal transduction of hormones may be related to spiral grain, as previously reported. *ARF7* and *MKK4_5*, as indoleacetic acid (IAA)- and ethylene (ET)-related receptors, may explain the contribution of plant hormones in spiral grain. This study provided useful information on spiral grain in *P. yunnanensis* by transcriptome and metabolome analyses and could lay the foundation for future molecular breeding.

## 1. Introduction

Spiral grain (as shown in Figure 1), a helical orientation around the axis of longitudinal tree trunks, is a common and negative wood property [1]. It occurs when tracheids deviate from the longitudinal axis of the trunk [2,3,4]. Spiral grain is mainly related to twisted types of wood, and spiral grain directly leads to twisting or warping, which reduces the strength of wood and limits wood products [5,6,7,8,9,10,11,12]. Trees with spiral grain have been reported in many pine species, including *Larix kaempferi* [13], *Pinus taeda* [14], *Pinus contorta* [15], *Pinus radiata* [16,17,18], *Picea abies* [3], *Picea sitchensis* [3], *Abies balsamea* [19,20], and *Pinus sylvestris* [20,21]. Most studies have suggested that spirality is highly heritable and that some environmental factors, such as wind [22], soil nutrient status [9], rainfall, slope, exposure [23,24], and altitude [25], have been proposed as possible causes of spiral grain.

The fibers in trees are normally thought of as directed along the axis of the tree. In reality, the fibers tend to be inclined in the tangential direction, thus forming a helix and spiral grain around the trunk axis [26]. This is a possible mechanism available for trees to improve environmental adaptation. The study showed that planting on sites exposed to strong prevailing winds [4] at wider spacing or undertaking heavy thinning can significantly increase the spiral grain angle and degree of twist. Furthermore, the plant hormone ethylene may function in the spiral grain growth of trees, although these suggestions have not been conclusively validated [20,27]. These studies were focused on environmental control, and few revealed the mechanisms based on molecular regulation.

Secondary metabolites are a class of compounds produced by organisms during growth and development and are closely related to their physiological functions and adaptations to the environment [28,29,30]. There is a close relationship between secondary metabolites and phenotype, and the type and content of secondary metabolites can directly affect the phenotypic characteristics of organisms [31]. For example, the secondary metabolites of some plants have biological activities such as antibacterial and antipest activities, which can protect plants from the external environment, thus affecting the growth and morphological characteristics of plants [31,32]. In addition, secondary metabolites can also affect the external manifestations of organisms, such as color [33], taste [34], and aroma [35], which, in turn, affect the reproduction [31] and adaptability [30] of organisms. The production and regulation of secondary metabolites are closely related to the phenotypic plasticity of organisms [36]. In the process of adapting to environmental changes, organisms produce different types of secondary metabolites in response to different stresses and challenges [30]; for instance, there are significant changes in the levels of antioxidant enzymes (such as superoxide dismutase (SOD), catalase (CAT), ascorbate peroxidase (APX), and guaiacol peroxidase (GPX)) in *Pisum sativum* seeds under salicylic acid and ultraviolet-B stresses [37]. The levels of phenolic pigments, carotenoids, and antioxidant enzymes differ in *Solanum lycopersicum* seeds following UV-B radiation and benzoic acid (BA) treatment, and all of the metabolites affect plant growth [38]. Some plants produce more antioxidants in response to oxidative damage when they are subjected to external environmental stresses, thereby altering their growth and morphological characteristics [30,39], such as *P. sativum* [37] and *S. lycopersicum* [40]. By modulating the production and regulation of secondary metabolites, organisms can adapt to environmental changes and alter their phenotypic characteristics, thereby enhancing their ability to survive and reproduce [41]. Thus, it is necessary to understand the metabolome of plant phenotypes.

*Pinus yunnanensis* is an important coniferous species in Southwest China and plays a critical role for the economy, society, and environment of Yunnan Province [42]. In nature forests, a number of *P. yunnanensis* trees are obviously characterized by spiral grain and withstand the negative impacts on timber performance and value. The characteristics of and differences in the transcriptome and metabolome between the straight- and twisted-trunk types of *P. yunnanensis* have not been investigated. In this study, integrated transcriptomic and metabolomic techniques were used to analyze and compare the differences in different spiral types of *P. yunnanensis*. This study aimed to determine the following: (1) that differential genes, metabolites, and pathways may be involved in different spiral types of *P. yunnanensis*; and (2) that metabolite characteristics are involved in the two trunk types of *P. yunnanensis*. The results of the present study add information on the molecular differential level in two different trunk types of *P. yunnanensis* and could lay the foundation for studies targeting the formation of spiral grain and further improving the wood quality of *P. yunnanensis*.

## 2. Results

### 2.1. De Novo Assembly

A total of 20 sample cDNA libraries generated an average of 46,789,947 raw reads (40,426,26~50,110,974) and 45,370,079 clean reads (39,685,518~48,414,246). With Q20 > 97.56% and Q30 > 93.91%, more than 7000 unigenes per sample were assembled, and the mean value of the unigenes was 826.55 bp for length, ranging from 787~858 bp and 1680.5 for N50 and ranging from 1637~1724. These indicated high-quality RNA-seq data.

### 2.2. Functional Annotation

Of 143,249 unigenes, 105,904 unigenes were annotated by at least one database, showing a 73.93% annotation rate (Appendix A). The abundance of unigenes was annotated by the non-redundant protein sequence (NR) (63.04%) and non-redundant nucleotide sequence (NT) (68.29%) databases. Over 70% of unigenes could be annotated with sequences from the top 7 hit species, of which 43,074 unigenes (47.7%) were assigned to *Picea sitchensis*. The E-value frequency distribution of hits (E-value ≤ 1.0 × 10^−60^) showed that 35.1% of the sequences shared strong homologies; the majority of matched sequences (64.9%) had E-values in the range of 1.0 × 10^−60^~1.0 × 10^−5^. The similarity values of most of the BLASTX hits (82.1%) were over 40% (Figure 2a–c).

The unigene sequences were searched against the Swiss-Prot protein databases using BLASTX. There were 62,994 unigenes annotated, of which 66.3% had E-values between 1.0 × 10^−50^ and 1.0 × 10^−5^, and 23.7% had strong homologies with E-values of ≤1.0 × 10^−50^.

In total, 56,932 unigenes were mapped to 127 Kyoto Encyclopedia of Genes and Genome (KEGG) pathways by the BLASTX server. Plant–pathogen interactions had the largest number of unigenes (12,059; 21.18%; ko01100), followed by plant hormone signal transduction (2918; 5.13%; ko04075), pyrimidine metabolism (2527; 4.44%; ko00240), purine metabolism (2449; 4.3%; ko00230), spliceosome (2136; 3.75%; ko03040), and RNA transport (2093; 3.68%; ko03013).

Clusters of orthologous groups of proteins (COGs) are an attempt to phylogenetically classify the complete complement of proteins (both predicted and characterized) encoded by complete genomes. Within the *P. yunnanensis* unigene set, 36,930 were categorized into 25 functional COG clusters (Figure 2d). The five largest categories, in order, are general function prediction, replication, recombination and repair, transcription, unknown function, and signal transduction mechanisms.

On the basis of the NR annotation, Blast2GO [43] was used to obtain a Gene Ontology (GO) annotation. Over 5000 unigenes were classified into the following three gene ontology categories: cellular component, biological process, and molecular function (Figure 2e). A large portion of unigenes consisted of the biological process category, in which most unigenes were assigned to the following five terms: cellular process, metabolic process, single-organism process, biological regulation, and response to stimulus. The terms in the cellular component category, including cell, cell part, organelle, and membrane part, presented large proportions. In the molecular function category, binding and catalytic activity were the top two terms annotated by most unigenes.

### 2.3. Differential Expression Gene Detection and Screening

Differences in gene expression between the straight and twisted trunk types were examined, and differentially expressed genes (DEGs) were identified by comparing the following unigene group pair expression changes: straight and twisted trunk types (S vs. T), secondary xylem and phloem (X vs. P), secondary xylem in straight and twisted trunk types (XS vs. XT), and phloem in straight and twisted trunk types (PS vs. PT). As shown in Figure 3a, a total of 6644 genes were significantly differentially expressed in the S vs. T pair, of which 3065 were up-regulated and 3579 were down-regulated. This indicated that a number of unigenes function in the straight trunk type. A total of 2115 unigenes were annotated by the KEGG database in the pair S vs. T. Moreover, there were obvious differences between X and P, and the DEG numbers were 13,418 (5797 up-regulated and 7620 down-regulated). Interestingly, 2066 DEGs (1026 up-regulated and 1040 down-regulated) and 1438 DEGs (555 up-regulated and 883 down-regulated) were found in the pair XS vs. XT and the pair PS vs. PT, respectively (Appendix A). The above results indicate that the number of down-regulated genes was greater in the twist trunk type than in the straight trunk type, regardless of the phloem or xylem of *P. yunnanensis*. This means that during the growth and development process, the expression of certain genes in the twist trunk type may be inhibited compared to that in the straight trunk type in *P. yunnanensis*. In addition, 783 and 581 were annotated by the KEGG database in XS vs. XT and PS vs. PT pairs, respectively. A radar diagram comparing the database annotation of DEGs in focus pairs was established (Figure 3b), and a large number of unigenes were assigned KEGG and GO databases in the pairs S vs. T, XS vs. XT, and PS vs. PT.

The results of an enrichment analysis showed that the plant–pathogen interaction was the most enriched pathway for the most differentially expressed genes in the three pairs. The pathways with significant differences were tyrosine metabolism and amino sugar and nucleotide sugar metabolism in the XS vs. XT pair. Glutathione metabolism, phenylpropanoid biosynthesis, and zeatin biosynthesis were the pathways of significant enrichment in the PS vs. PT pair, and plant hormone signal transduction was the second most abundant pathway for gene enrichment in the PS vs. PT pair. Above all, plant–pathogen interactions were attributed to the occurrence and development of spiral grains in *P. yunnanensis*, and spiral grains can be affected by glutathione metabolism, phenylpropanoid biosynthesis, zeatin biosynthesis, plant hormone signal transduction, and other pathways (Figure 4).

### 2.4. Metabolite Characteristics of P. yunnanensis

This study only used phloem as the sample because phloem plays a more important role in plant growth and development than xylem. Metabolites were detected using an UHPLC-QTOF-MS analysis [44], and a total of 2447 metabolites were detected in *P. yunnanensis*. An orthogonal partial least-squares discrimination analysis (OPLS-DA) demonstrated a clear separation between the PT group and PS group (Appendix A). A total of 126 differentially accumulated metabolites (DAMs) were detected. Among them, 60 were from the positive mode and 69 were from the negative mode. Ent-epicatechin (4alpha->8) catechin, quercetin 3-(2″-p-hydroxybenzoyl-4″-p-coumarylrhamnoside, glycyl-aspartate, 1-heptadecanoyl-sn-glycero-3-phosphocholine, and PC (16:0/0:0) were the top five down-regulated DAMs in the positive mode, and 6-methyladenine, glucosaminic acid, gluconolactone, sodium gluconate, and 5-methylcytidine were the top five DAMs in the negative mode (Figure 5). Moreover, three metabolites (l-saccharopine, 5-methylcytidine, and 13-OxoODE) were common between the positive mode and the negative mode. Most of the metabolites accumulated more in the positive mode in the PS group, except for 3 metabolites (tremetone, 10-hydroxycamtothecin, and 4-acetamidobutanoate), and 44 metabolites accumulated more in the negative mode in the PT group, e.g., geldanamycin, novobiocin, and matairesinol (Appendix A). The DAMs were further annotated with class-specific terms from seven categories (Appendix A) via a search on the human metabolome database (HMDB), including carboxylic acids and derivatives, fatty acyls, benzene and substituted derivatives, pyrimidine nucleosides, organooxygen compounds and indoles, and derivatives. These were assigned to five classes, including the following eight terms in the Lipid Map database: steroids (epitestosterone), isporenoids (ginkgolide C), linear tetracyclines (oxytetracycline), macrolides and lactone polyketides (zearalenone), flavonoids (procyanidin B2, quercetin 3-(2″-p-hydroxybenzoyl-4″-p-coumarylrhamnoside), epicatechin 3-O-beta-D-allopyranoside, 6′-Hydroxy-4,2′,3′,4′-tetramethoxychalcone, calomelanol A, and butein), glycerophosphocholines (1-heptadecanoyl-sn-glycero-3-phosphocholine), fatty acyls and conjugates (8,11-Octadecadiynoic acid and sebacic acid), and octadecanoids (13-HOTE and 13-OxoODE) (Appendix A). By annotating with the KEGG database, 31 metabolites were assigned to 71 metabolic pathways, including carbon metabolism, tryptophan metabolism, caffeine metabolism, fructose and mannose metabolism, glycine, and serine and threonine metabolism (Appendix A).

### 2.5. Gene Coexpression Network Construction

Twenty-four differential pathways were common in the transcriptome and metabolome, which were mainly annotated as ABC transporters, arginine and proline metabolism, flavonoid biosynthesis, isoquinoline alkaloid biosynthesis, linoleic acid metabolism, phenylpropanoid, tryptophan metabolism, etc. However, there was no clear pathway constructed for gene expression and other metabolite abundance in the spiral grain group. To further explore the important factors involved in spiral grain formation, the correlations of 24 pathways, 23 metabolites, and 72 unigenes (Appendix A) were analyzed (Appendix A). A total of 16 metabolites were highly correlated with unigenes, which were selected as the candidate traits for further analysis.

Coexpression networks were analyzed using 21,558 annotation unigenes. These unigenes were clustered into 18 modules labeled with different colors, and each module contained unigenes with similar expression patterns (Figure 6a). Among the 18 modules, lightblue4 was the only one that was highly correlated with the traits (r > 0.8; *p* < 0.05). The lightblue4 module contains 390 unigenes, which include 21 TFs (bZIP, HD-ZIP, bHLH, C3H, NAC, etc.). A heatmap of module–trait relationships showed that the lightblue4 module highlighted the metabolites that differed significantly between the straight and twisted trunks of *P. yunnanensis*. For example, the abundance of 2′-deoxyuridine was higher in the PS group. A GO analysis of the lightblue4 module showed that its unigenes were significantly enriched in glucose transport, monosaccharide transmembrane transport, and hexose transmembrane transport in the biological process category and significantly enriched in active transmembrane transporter activity, symporter activity, and sugar proton symporter activity in the molecular function category (Appendix A). A KEGG enrichment analysis revealed that glutathione metabolism, starch and sucrose metabolism, and cyanoamino acid metabolism were most strongly enriched in the lightblue4 module genes (Appendix A). A cytoscape interaction network showed the interaction of TFs and two metabolites (2′-deoxyuridine and butein) (Figure 6b), which indicated that the TFs may be closely related to the spiral grain of *P. yunnanensis*.

### 2.6. RT-qPCR Verification

Based on a DEG analysis and the coexpression network, a total of ten unigenes were used to validate the transcriptome results, which included genes related to plant hormone signal transduction (*CL3988.Contig2_All* (*MKK4_5*), *CL7342.Contig1_All* (*BAK1*), *Unigene31619_All* (*ARF7*), *CL2039.Contig1_All* (*WRKY33*), *CL12088.Contig3_All* (*COI1*), *CL1065.Contig7_All* (*CRE1*), and *CL5455.Contig3_All* (*PIF3*)), transporters (*Unigene43064_All* (*ABCG2*) and *Unigene28800_All* (*ABCC2*)), and microtubule-related proteins *CL4761.Contig5_All* (*KIF11*). The RT-qPCR analyses showed trends in expression that were consistent with those found by RNA-seq (Appendix A), suggesting that the transcriptome results were reliable.

## 3. Discussion

The cell wall, a major structural determinant of plants, can regulate the extent and symmetry of plant cell growth and is a complex matrix of polysaccharides and proteins. The polysaccharides comprise cellulose, hemicellulose, and pectin [45]. Cellulose synthase (CesA), organized in the plasma membrane, can synthesize cellulose [46]. Once CesA combines with 18 glucose chains, cellulase microfibrils are formed [47]. The direction of cell expansion depends on the angle of the cellulose microfibrils, with maximal cell expansion typically perpendicular to the net orientation of the cellulose microfibrils [48]. The patterns of cellulose microfibrils are disturbed by microtubules [49]. SPIRAL1, a microtubule-based modulator, is one of the first proteins identified that determines the twisted growth of plants [50,51]. In addition, SPIRAL1, *ark2* [52], *csi1*/*pom2* [53], *eb1* (*a*,*b*,*c*) [54], *mor1* [55], *spr1*/*sku6* [56], and *tor1*/*spr2* [57], which encode microtubule-associated proteins, have been proven to play a role in the helical growth of *A. thaliana*. In this study, two differential microtubule-associated proteins (AURKX and KIF11) were also found in straight- and twisted-trunk *P. yunnanensis*. Therefore, we speculated that these two proteins may also be one of the factors affecting the formation of twisted-trunk *P. yunnanensis*. Of course, their functions need further analysis. In addition, previous research has also found that cell-wall-related genes (e.g., *rhm1*, *cob*, and *sku5*) play a role in spiral grain [58,59,60]. It has been demonstrated that helical twist is related to cell expansion and that the helical pattern of cell expansion is associated with altered cell wall composition [58]. The cell wall composition could be directly or indirectly regulated by plant hormones. For example, gibberellinin directly induces XET expression in rice leaf sheaths [61]. The walls of GA_3_-treated Hong Mang Mai seedlings showed increased extensibility, which resulted in changes in wall glucan content, autolysis, and glucanase activity [62].

The plant hormone ethylene plays a key role in the rate of cambial division, tracheid cell wall biochemistry, and tracheid morphology and may function in the formation of spiral grains. In an earlier study, an increased number of early wood tracheids in trunk sections was correlated with a more left-handed spiral grain angle, and increased ethylene (ET) regulates the extent of the spiral grain angle [20]. Indole-3-acetic acid (IAA) has been shown to participate in the regulation of spiral grain formation [27]. These findings indicate that these two plant hormones are involved in the quantitative regulation of wood formation [20,63]. In this study, 27 DEGs were associated with plant hormone signal transduction. They encode the following 11 compounds from 7 hormone signaling pathways (Figure 7): ARF (auxin response factor) in auxin (IAA) signaling; CRE1 (cytokinin receptor) in cytokinin (CTK) signaling; GID1 (gibberellin receptor GID1) and TF (phytochrome-interacting factor 4, PIF4) in gibberellin (GA) signaling; PYR/PYL (abscisic acid receptor PYR/PYL family, PYL) in abscisic acid (ABA) signaling; SIMKK (mitogen-activated protein kinase 4/5, MKK4/5) in ethylene (ET) signaling; BAK1 (BRI1-associated receptor kinase 1), BRI1 (brassinosteroid-insensitive 1), and BSK (BR signaling kinase) in brassinosteroid (BR) signaling; and COI1 (coronatine-insensitive 1) and JAZ (jasmonate ZIM domain-containing protein) in jasmonic acid (JA) signaling. Surprisingly, few DAMs were detected in plant hormone signal transduction, which may be transported to be utilized in others. These results, to some extent, indicated that plant hormones may contribute to the formation of spiral grains in *P. yunnanensis*.

Environmental factors largely contribute to the developmental growth of plants. Many studies have focused on the relationship between spiral grain and abiotic factors and hypothesized that the formation of spiral grain is induced by multiple external stresses. In addition, no clear molecular explanation is obtained for the spiral grain occurrences. A large number of DEGs (416 unigenes, mainly coding 14 proteins) between the straight type and twist type of *P. yunnanensis* were mapped to the plant–pathogen interaction pathway, including CPK (calcium-dependent protein kinase), CML (calcium-binding protein CML), FLS2 (LRR receptor-like serine/threonine protein kinase FLS2), RPM1 (disease resistance protein RPM1), and WRKY33 (WRKY transcription factor 33). Meanwhile, some DAMs for terpenoid compounds (ginkgolide C and nomilin), microbial metabolism in diverse environments (d-lactic acid, (S)-2-hydroxyglutarate, paraxanthine, 4-methylcatechol, and glucosaminic acid), antimicrobials (oxytetracycline, sulfamethopyrazine, nafcillin, and norfloxacin), and natural toxins (zearalenone) were found. All of these metabolites are involved in the defense of plants against both biotic and abiotic responses [64]. The nucleoside derivative 2′-deoxyuridine plays a vital role in DNA replication and repair processes within living organisms. Its significance lies in its ability to act as a substrate for DNA synthesis enzymes, contributing to the maintenance of genetic stability [65]. In addition, the study found that the rhizosphere microbial community between the straight and twisted trunk types of *P. yunnanensis* was different [66]. In summary, the differences in microbes may affect the growth of *P. yunnanensis* based on plant–microbe interactions. Unlike mammals, plants have no mobile defender cells and a somatic adaptive immune system [67]. Instead, they rely on two distinct types of immune receptors: pattern-triggered immunity (PTI), which is based on the recognition of molecules common to many classes of microbes, and effector-triggered immunity (ETI), which is based on the response to pathogen virulence factors on host targets [68,69,70]. PTI is initiated by pathogen-associated molecular patterns (MAMPs/PAMPs) with pattern recognition receptors (PRRs), including FLS2, which forms a receptor-like kinase complex with BAK1 and recognizes a conserved 22-amino-acid fragment (flg22) of bacterial flagellin [70,71]. Furthermore, ETI is induced through the recognition of pathogen effectors by plant disease resistance (R) proteins, including RPM1 and RPS2. In this study, these proteins were encoded by most DEGs and were identified as important hubs for changes in the plant–pathogen interaction pathway. Consequently, the alteration of hub protein activities in twisted-trunk *P. yunnanensis* has a large influence on plant immunity, which is highly related to the spiral grain growth of trees.

## 4. Materials and Methods

### 4.1. Plant Materials

Based on similar diameters and tree heights, five natural *P. yunnanensis* trees with straight trunks and twisted trunks (spiral angle > 30°) were chosen, respectively. All trees were selected in the nature forest on Mount Fangshan (E 101°46′24″, N 26°7′08″), Yongren County, Chuxiong Yi Autonomous Prefecture, Yunnan Province, China. The group of *P. yunnanensis* samples was mainly defined as follows: twisted xylem was XT, straight xylem was XS, twisted phloem was PT, and straight phloem was PS. For each group, we collected samples from five trees, which yielded five biological replicates for RNA-seq analyses. In addition, we also artificially divided the plants into four other groups, namely, the twisted group (T, XT+XP), straight group (S, PT+PS), xylem group (M, XT+XS), and phloem group (P, PT+PS) for a follow-up analysis and PT and PS for a metabolome analysis.

### 4.2. RNA Isolation and cDNA Library Construction

Total RNA from 20 samples was isolated using the RNAprep pure plant kit (Tiangen Biotech, Beijing, China). The RNA concentration was determined using a spectrophotometer at A260/280 (NanoDrop 1000; Thermo Scientific, Wilmington, DE, USA), and the integrity was assessed with an Agilent 2100 system (Agilent Technologies, Santa Clara, CA, USA). The qualified RNA samples were used for complementary DNA (cDNA) library construction. Each library produced approximately 6 gigabases (Gb) of raw data on the Hiseq-PE150 platform. The transcriptome data in this study were submitted to the NCBI short-read archive (SRA) database under accession number PRJNA507489.

### 4.3. RNA-Seq Analysis

The readings were assembled using Trinity [72]. In brief, first, a k-mer (k = 25) dictionary was constructed from all sequence reads, and Inchworm was used to assemble the read dataset and generate linear contigs. Then, the Chrysalis clusters related the contigs that comprise alternatively spliced transcripts or other paralogous genes. Finally, Butterfly analyzed the paths of consecutive plausible sequences and generated longer transcript sequences. The result of Trinity’s assembly is shown as unigenes. The final unigenes were obtained by clustering the homologous transcript sequences using Tgicl version 2.1 [73]. All unigenes assembled and spliced from the 20 final unigenes were submitted to BLASTX [74] alignment (E-value of less than 10^−5^) in NT, NR, Swiss-Prot, COG, GO, KEGG, PlantTFdb (plant transcription factor database), PRGdb (plant resistance gene database), InterPro, and STRINGdb (a search tool for the retrieval of interacting genes/proteins database), respectively. These databases were used to estimate unigene functions.

DEGs between pairs of sample groups were detected using the DEGseq2 package [75] based on read count. The parameters for DEG detection were as follows: (1) *p* value ≤ 0.05 and (2) |log_2_FC| ≥ 1. The DEGs were further classified into the following two patterns: up-regulation and down-regulation. TBtools [76] was used to search the enriched GO terms and KEGG pathways, and the visualization was carried out with the ggplot2 package [77].

### 4.4. Metabolome Sample Preparation and Analyses

The phloem sample (50 mg) was extracted with 1000 μL of extract (the volume proportion of methanol, acetonitrile, and water was 2:2:1) and 2 μL of 2-chlorol-L-phenylalalnine (2 mg/L) (as an internal quantitative standard). All samples were vortexed and mixed for 30 s, then ground into a fine powder using a grinding mill at 45 Hz for 10 min, and ultrasonicated for 10 min in an ice bath. After resting at −20 °C for 1 h, the samples were centrifuged at 12,000 rpm and 4 °C for 15 min, and 500 μL of supernatant was transferred to EP tubes, and the dried extracts were concentrated under vacuum. Next, 160 mL of the acetonitrile and water extract (the volume proportion was 1:1) was added to the dried extract and redissolved, vortexed for 30 s, ultrasonicated for 10 min in an ice bath again, and centrifuged at 12,000 rpm and at 4 °C for 15 min again. Finally, 120 mL of supernatant was transferred to a vial for UHPLC-QTOF-MS analysis. Metabolites were detected using the UHPLC-QTOF-MS analysis platform (Waters Acquity I-Class PLUS UHPLC and Waters Xevo G2-Xs QTof HRMS) at Biomarker Technologies Corporation (Beijing, China). Chromatographic analysis was performed on an Acquity UPLC HSS T3 column (Waters; 100 mm × 2.1 mm, 1.8 μm). The mobile phases comprised A (water containing 0.1% formic acid) and B (0.1% formic acid in acetonitrile). The gradient elution program was as follows: 2% B at 0~0.25 min, 2~98% B at 0.25~10.0 min, 98% B at 10.0~13.0 min, 2~98% B at 13.0~13.1 min, and 2% B at 13.1~15.0 min. The injection volume was 1 μL and eluted at a flow rate of 0.4 mL/min. The acquisition rate was set at 5 spectra/s. Metabolites were acquired under 2.0 kV (positive ion mode) and 1.5 kV (negative ion mode) capillary voltages. The parameters of MS data acquisition were set as follows: a cone voltage of 30 V, an ion source temperature of 150 °C, a desolvent gas temperature of 500 °C, and a backflush gas flow rate and desolventizing gas flow of 50 L/h and 800 L/h, respectively. OPLS-DA was carried out to compare the differences in the spiral grain group using the ropls packages of R (3.3.2 version). The parameters VIP (variable importance in the projection) > 1 and *p* < 0.05 were the basis for DAM identification. Annotation of metabolites was performed using databases including the HMDB, KEGG, and Lipid Maps databases.

### 4.5. Coexpression Network Construction

A coexpression network analysis (WGCNA) [78] of approximately 22,000 genes was performed using the R package based on the unigenes with annotation. The common pathways with DEGs and DAMs in PS vs. PT were screened, and a correlation analysis was performed between DEGs and DAMs. The abundance of DAMs was used as the phenotypic traits file. Modules highly correlated with important metabolites were identified. The expression network of unigenes in the highlighted module was performed for visualization using Cytoscape 3.6.1 [79].

### 4.6. Quantitative Real-Time Polymerase Chain Reaction

To validate the results of the expression profiling obtained by RNA-seq, RT-qPCR (real-time quantitative PCR) was performed to compare ten DEGs between two different trunk types (Appendix A). The total RNA isolation RNAprep pure plant kit (Tiangen Biotech, Beijing, China, DP441) was employed to extract RNA from different trunk types and tissues of *P. yunnanensis*. The RNA was checked using a Nanodrop (NanoDrop 1000; Thermo Scientific, Wilmington, DE, USA) and an Agilent 2100 Bioanalyzer (Agilent Technologies), and the qualified RNA was used as a template for cDNA synthesis. Reverse transcription was performed using the FastQuant RT kit (Tiangen Biotech, Beijing, China). Chemical fluorescence dry (ChamQ SYBY qPCR Master Mix; Nanjing, China) was used to check the expression of hub genes or DEGs on a real-time PCR instrument (Rotor-Gene Q; Qiagen, Germany). The reference gene *AT2G24020* from *Pinus pinaster* [80] was used to quantify the relative mRNA levels. The primer pairs were designed using Primer Premier 5 [81] and synthesized by Sangon Biotech Co., Ltd. (Shanghai, China). A three-step amplification was carried out as follows: 95 °C for 2 min; 45 cycles at 95 °C for 10 s, 56 °C for 30 s, and 72 °C for 60 s. The 2^−∆∆Ct^ method [82] was used to calculate the relative expression level. This RT-qPCR was performed using five biological replicates.

## 5. Conclusions

Many naturally occurring *P. yunnanensis* trees are characterized by spiral grain, which limits their importance in forestry production, afforestation, and environmental conservation. Previous studies hypothesized that the formation and growth of spiral grains in *P. yunnanensis* were induced by some abiotic and biotic factors. However, the regulatory mechanisms of metabolites and genes, as well as their correlation, are unclear. Integrative transcriptome and metabolome techniques were used to investigate how spiral grains formed in *P. yunnanensis*. A transcriptome analysis identified that ARF7 and MKK4_5, the IAA- and ET-related receptors, may explain the contribution of plant hormones in spiral grain. Differences in transporters (ABCG2 and ABCC2) were found in this study, which indicated that the difference in spiral grain may be affected by nutrient or water availability. Resistance proteins (RPM1 and RPS2) were also significantly different in the straight and twisted trunk types. A gene enrichment analysis revealed an important role of the plant–pathogen interaction pathway in the growth of spiral grains. Based on the metabolome, terpenoid compounds (ginkgolide C and nomilin) were differentially abundant in the two different trunk types of *P. yunnanensis*. Additionally, a correlation analysis identified that transcription factors such as bZIP, HD-ZIP, bHLH, C3H, and NAC were strongly associated with 2′-deoxyuridine and butein, of which 2′-deoxyuridine indirectly participates in the synthesis of terpenoids. The above results indicated that plant hormones, plant immunity, and terpenoid compounds exert strong effects on the formation and growth of spiral grains in *P. yunnanensis*. In summary, the excavation of transcriptomic and metabolomic resources, as well as the establishment of plant metabolic regulation models, can facilitate the exploration of the mechanism of spiral grain formation in *P. yunnanensis*.

Above all, the present study concludes that plant cell walls, hormones (IAA and ET), and secondary metabolites (terpenes and nucleoside derivatives) are of interest in the further study of spiral grain in trees.

## Figures and Tables

**Figure 1 ijms-24-14658-f001:**
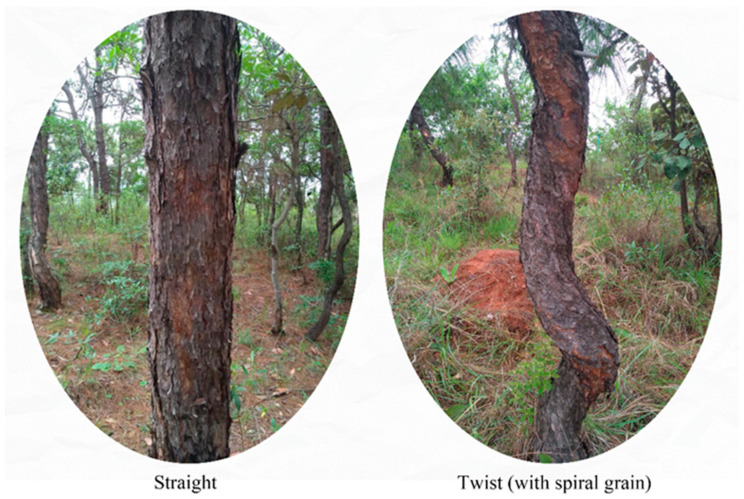
Spiral grain in *Pinus yunnanensis*.

**Figure 2 ijms-24-14658-f002:**
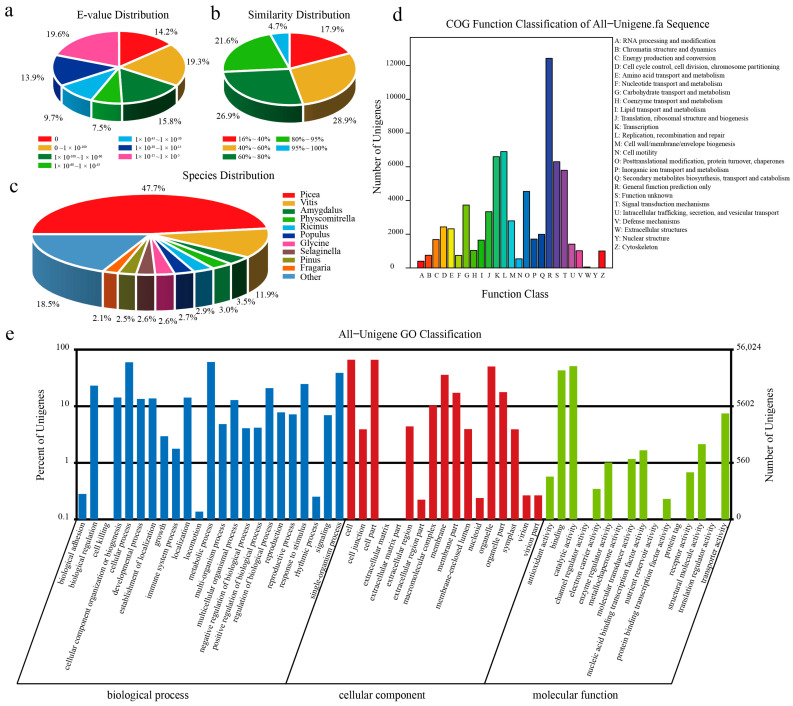
The annotated result of NR, COG, and GO. (**a**) NR of E-value distribution. (**b**) NR of similarity distribution. (**c**) NR of species distribution. (**d**) COG function classification. (**e**) GO classification.

**Figure 3 ijms-24-14658-f003:**
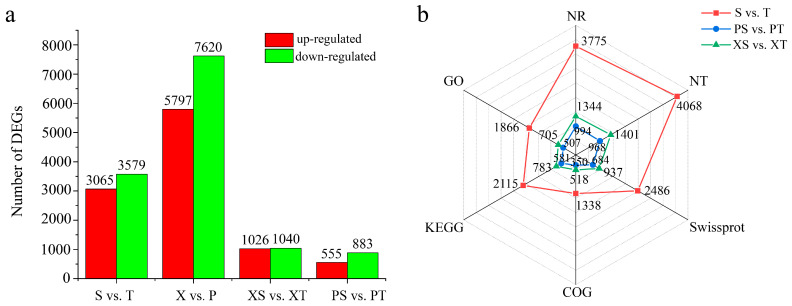
Information on DEGs. (**a**) DEG counts in different pairs. (**b**) Annotation of DEG numbers by NR, NT, Swissport, COG, KEGG, and GO databases in different pairs. S: straight; T: twisted; X: xylem; P: phloem; XS: straight xylem; XT: twisted xylem; PS: straight phloem; PT: twisted phloem.

**Figure 4 ijms-24-14658-f004:**
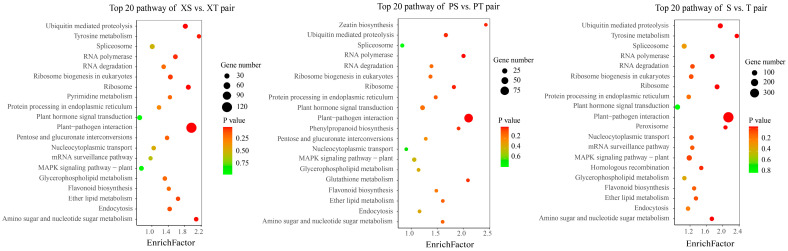
KEGG enrichment of XS vs. XT, PS vs. PT, and S vs. T pairs.

**Figure 5 ijms-24-14658-f005:**
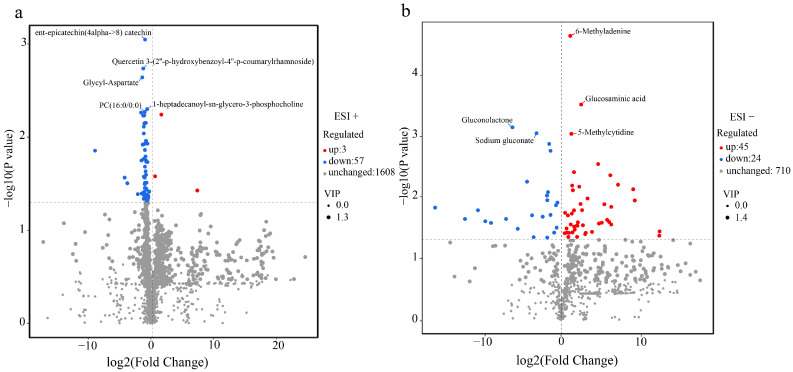
DAM screening in the PS vs. PT pair. (**a**) DAM volcano in positive mode (ESI+); (**b**) DAM volcano in negative mode (ESI−).

**Figure 6 ijms-24-14658-f006:**
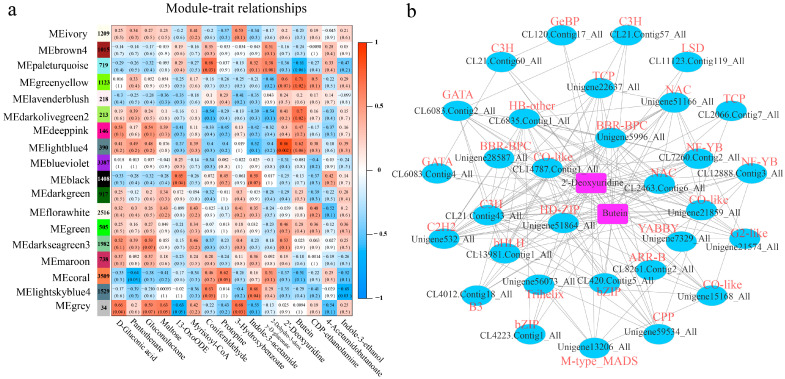
WGCNA analysis and hub unigene search. (**a**) Relationships between gene modules and metabolite traits. (**b**) Network of transcription factors (TFs) in modules highly connected to 2′-deoxyuridine and butein metabolites. Each ellipse node indicates a TF gene, which is labeled in red, and the round rectangle node indicates a metabolite; they are labeled in blue.

**Figure 7 ijms-24-14658-f007:**
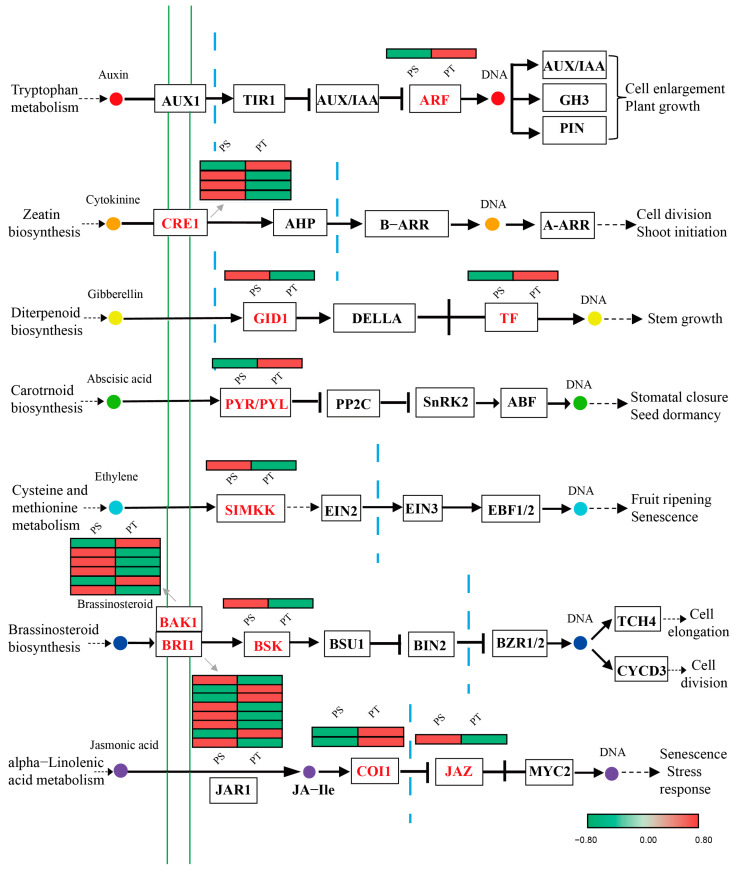
Differentially expressed genes assigned to plant hormone signal transduction. The map is from the Kyoto Encyclopedia of Genes and Genomes (KEGG). The gene expression levels of each treatment are shown in different columns, as described in the color key, and the different rows represent the different genes.

## Data Availability

The transcriptome data in this study are available from the NCBI SRA database under accession number PRJNA507489.

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
