# Peer review of "Comparative Transcriptomic and Metabolomic Analyses of Differences in Trunk Spiral Grain in Pinus yunnanensis"

_ijms, 2023, doi:10.3390/ijms241914658_

Round 1
Reviewer 1 Report (Previous Reviewer 2)
The manuscript has been revised accordingly and I don't have further questions.
Author Response
Thanks for your suggestions.
Reviewer 2 Report (New Reviewer)
The authors tried to understand metabolic and molecular level differences between Pinus yunnanensis trees with strait trunk and twisted trunk. The work was well planned and the authors presented good results. However, many places are not very understandable. It requires a neat language correction. For example, the second line in abstract “Although some hypothesis around the abiotic factor have been proposed, none adequately molecule explains is obtained for the occurrences”, which is very confusing. Please check the whole manuscript for such errors. Apart from this please check the following comments also,
1. It’s better to avoid start a sentence with number. For example “24 common differential pathways, were from transcriptome and metabolome” may be written as “There were 24 common differential pathways identified from the transcriptome and metabolome”.
2. Expansions of any abbreviations used in abstract is needed (Eg. IAA and ET need to be expanded).
3. The authors mention 20 cDNA was synthesised. However, they also mentioned that 20 samples from 6 different groups with 5 biological replications were collected. Whether they pooled all 5 biological replications before RNA extraction? How these 20 samples represent the 6 different categories? Also, in the methods (in section 4.1), after they mentioned 6 groups, they explained totally 8 groups (XT, XS, PT, PS, T, S, X & P). Clarify this.
4. HPLC methods which was used is missing in the materials section.
5. Although primer information is available in the supplementary file, it was not numbered and represented in the text.
The manuscript is not understandable in many places. Stringent grammar checking and language editing are required
.
Author Response
Comments and Suggestions for Authors
The authors tried to understand metabolic and molecular level differences between Pinus yunnanensis trees with strait trunk and twisted trunk. The work was well planned and the authors presented good results. However, many places are not very understandable. It requires a neat language correction. For example, the second line in abstract “Although some hypothesis around the abiotic factor have been proposed, none adequately molecule explains is obtained for the occurrences”, which is very confusing. Please check the whole manuscript for such errors. Apart from this please check the following comments also,
- It’s better to avoid start a sentence with number. For example “24 common differential pathways, were from transcriptome and metabolome” may be written as “There were 24 common differential pathways identified from the transcriptome and metabolome”.
Thanks for your suggestions, we check the whole manuscript for such errors and correct these.
- Expansions of any abbreviations used in abstract is needed (Eg. IAA and ET need to be expanded).
We sincerely appreciate your suggestions, all abbreviations used in abstract where have been expanded.
- The authors mention 20 cDNA was synthesised. However, they also mentioned that 20 samples from 6 different groups with 5 biological replications were collected. Whether they pooled all 5 biological replications before RNA extraction? How these 20 samples represent the 6 different categories? Also, in the methods (in section 4.1), after they mentioned 6 groups, they explained totally 8 groups (XT, XS, PT, PS, T, S, X & P). Clarify this.
We sincerely appreciate your suggestions, we didn't give clear statement about every type of group. The group of P. yunnanensis sample are mainly defined as follows: xylem of twisted type was XT, xylem of straight type was XS, phloem of twisted type was PT, phloem of straight type was PS. For each group, we collected samples from five trees, which yielded five biological replicates for RNA-seq analysis. In addition, we also artificially divided into four other groups, namely the twisted group (T, XT+XP), straight group (S, PT+PS), xylem group (M, XT+XS) and phloem group (P, PT+PS) for follow-up analysis. PT and PS for metabolome analysis.
- HPLC methods which was used is missing in the materials section.
We sincerely appreciate your suggestions, HPLC methods were add into the 4.4 section.
- Although primer information is available in the supplementary file, it was not numbered and represented in the text.
We sincerely appreciate your suggestions, label the primer information in the text.
Please see the attachment

Reviewer 3 Report (New Reviewer)
Overall Summary: The study by Gan et al. examines the differences between straight and spiral growth orientations of Pinus yunnanensis using a combination of transcriptomics and metabolomics. They evaluate the global patterns generated by these two approaches and conclude that plant hormones and secondary metabolites contribute to this physiological differentiation. The manuscript provides valuable statistical analysis of the multiomic datasets. However there are issues with the presentation of the data, the resolution of the figures presented and English language use. Specific comments are provided below.
Specific Comments:
· The manuscript needs to be evaluated for grammar and sentence construction issues throughout. There are issues with the use of prepositions and the tense of the sentences shifting midway. These need to be thoroughly evaluated and corrected.
· Page 2, 2nd Paragraph: the introduction for secondary metabolites is too general and does not include any direct statements about the role of secondary metabolites in particular organisms.
· Methods 4.1: It is not clear what ‘from nature’ means. Provide specific details on the forest, including names of locations and co-ordinates of where the samples were collected from.
· There need to be images that represent what XT, PT, S and other types of samples mean.
· Figure 2: The resolution of this figure is very low. It is not possible to read the legends on the figure at all. This needs to be corrected and the font size needs to be increased and the blurriness of the image should be reduced.
· Figure 4, 5 and 6 also do not have good resolution. It is not possible to read the font.
· A list of the top 25/50 genes that are up and down regulated in each of the comparative conditions, along with P-values and log fold change in expression should be included in the main manuscript to indicate the major impact in terms of the transcriptome. This should also be described in detail in the results and discussion section with specific details on the potential patterns of regulation indicated by the top up and down regulated genes and the physiological differences between the sample types.
· Figure 5: A and B. The volcano plots should have different metabolites labelled within the plot for some of the dots, especially the ones that are outliers.
· What was the reason for the metabolomic profiling not including all the different sample types used in RNA seq profiling. This needs to be explained in the results section.
· The sample types chosen for metabolomic analysis are not all the ones used in RNAseq, if data is present about other samples it should be included in the study, if not, the distinction between the sample types chosen for RNASeq and metabolomics should be explained.
The manuscript needs to be evaluated for grammar and sentence construction issues throughout. There are issues with the use of prepositions and the tense of the sentences shifting midway. These need to be thoroughly evaluated and corrected.
Author Response
Dear Reviewers,
Thank you very much for your time involved in reviewing the manuscript and your very encouraging comments on the merits.
Comments:
“The study by Gan et al. examines the differences between straight and spiral growth orientations of Pinus yunnanensis using a combination of transcriptomics and metabolomics. They evaluate the global patterns generated by these two approaches and conclude that plant hormones and secondary metabolites contribute to this physiological differentiation. The manuscript provides valuable statistical analysis of the multiomic datasets. However, there are issues with the presentation of the data, the resolution of the figures presented and English language use. Specific comments are provided below.”
We also appreciate your clear and detailed feedback and hope that the explanation has fully addressed all of your concerns. In the remainder of this letter, we discuss each of your comments individually along with our corresponding responses.
To facilitate this discussion, we first retype your comments in italic font and then present our responses to the comments.
Comments 1:
The manuscript needs to be evaluated for grammar and sentence construction issues throughout. There are issues with the use of prepositions and the tense of the sentences shifting midway. These need to be thoroughly evaluated and corrected.
Response 1:
Thank you for the detailed review. We have carefully and thoroughly proofread the manuscript to correct the grammar and typos.
We have the article
Comments 2:
2nd Paragraph: the introduction for secondary metabolites is too general and does not include any direct statements about the role of secondary metabolites in particular organisms.
Response 2:
We sincerely appreciate your suggestions. Reference for ‘secondary metabolites’ play role in particular organisms have added into the Introduction section.
Comments 3:
Methods 4.1: It is not clear what ‘from nature’ means. Provide specific details on the forest, including names of locations and co-ordinates of where the samples were collected from.
Response 3:
We sincerely appreciate your suggestions, We have adjusted the description about ‘Plant materials’ section, ‘All trees were selected in nature forestry from mount Fangshan (E 101â—¦46’24’’, N 26°7′08″), Yongren county, Chuxiong Yi Autonomous Prefecture, Yunnan province, China.
Comments 4:
There need to be images that represent what XT, PT, S and other types of samples mean.
Response 4:
We sincerely appreciate your suggestions, we didn't give clear statement about every type of group. The group of P. yunnanensis sample are mainly defined as follows: xylem of twisted type was XT, xylem of straight type was XS, phloem of twisted type was PT, phloem of straight type was PS. For each group, we collected samples from five trees, which yielded five biological replicates for RNA-seq analysis. In addition, we also artificially divided into four other groups, namely the twisted group (T, XT+XP), straight group (S, PT+PS), xylem group (M, XT+XS) and phloem group (P, PT+PS) for follow-up analysis. PT and PS for metabolome analysis.
Comments 5:
Figure 2: The resolution of this figure is very low. It is not possible to read the legends on the figure at all. This needs to be corrected and the font size needs to be increased and the blurriness of the image should be reduced.
Response 5:
We sincerely appreciate your suggestions, the Figure 2 is too many the element, and we will try ours best to adjust it, if not, whether we can put these to “attachment figure”, such as divide into “attachment Figure 2a”, “attachment Figure 2b”, “Figure 2c” and so on, respectively.
Comments 6:
Figure 4, 5 and 6 also do not have good resolution. It is not possible to read the font.
Response 6:
We sincerely appreciate your suggestions, the Figure 4, 6 is too many the element, and we will try ours best to adjust it, if not, whether we can put these to “attachment figure”. Figure 5 is hard to adjust the font to read, we decide to put the Figure 5c, 5d, the two heatmap to the attachment.
Comments 7:
A list of the top 25/50 genes that are up and down regulated in each of the comparative conditions, along with P-values and log fold change in expression should be included in the main manuscript to indicate the major impact in terms of the transcriptome. This should also be described in detail in the results and discussion section with specific details on the potential patterns of regulation indicated by the top up and down regulated genes and the physiological differences between the sample types.
Response 7:
Thanks for your suggestions, we add the list of the all genes in each of the comparative conditions, and make it description in text.
Comments 8:
Figure 5: A and B. The volcano plots should have different metabolites labelled within the plot for some of the dots, especially the ones that are outliers.
Response 8:
We sincerely appreciate your suggestions, but the element is too many, we will label the top 5 the different metabolites, and make it description.
Comments 9:
What was the reason for the metabolomic profiling not including all the different sample types used in RNA seq profiling. This needs to be explained in the results section.
Response 9:
We sincerely appreciate your suggestions. We will explain it in the results section.
Comments 10:
The sample types chosen for metabolomic analysis are not all the ones used in RNAseq, if data is present about other samples it should be included in the study, if not, the distinction between the sample types chosen for RNASeq and metabolomics should be explained.
Response 10:
We sincerely appreciate your suggestions, metabolomic sampling mainly uses the phloem rather than the xylem, as the phloem plays a more important role in plant growth and development. Moreover, xylem sampling is difficult, and our sample size is not sufficient to complete metabolomic analysis.
Dear Reviewer:
We would like to take this opportunity to thank you for all your time involved and this great opportunity for us to improve the manuscript. We hope you will find this revised version satisfactory.
Yours sincerely,
Peihua Gan
Please see the attachment.

Round 2
Reviewer 2 Report (New Reviewer)
Although the authors completed the revision, still some corrections are required.
1. The authors still didnot numbered the primer table present in the supplementary file. It was not mentioned in the methods section also. (10 unigenes primer information in supplementary folder).
2. Abbreviating botanical name in the second or lateral usagae is missed in some places (Eg. In third para of Introduction line number 22 "Solanum lycopersicum".
3. Check the fonts. In section 4.6 rt-QPCR or RT-qPCR. use uniform style; similarly "template fOR CDNA synthesis" in line 7 as "template for CDNA synthesis".
Minor corrections are required
Author Response
- The authors still didnot numbered the primer table present in the supplementary file. It was not mentioned in the methods section also. (10 unigenes primer information in supplementary folder).
We sincerely appreciate your suggestions, 10 unigenes primer information as the “Table S5. Primer information” have been numbered in methods section.
- Abbreviating botanical name in the second or lateral usagae is missed in some places (Eg. In third para of Introduction line number 22 "Solanum lycopersicum".
We sincerely appreciate your suggestions, we check and corrected the such error.
- Check the fonts. In section 4.6 rt-QPCR or RT-qPCR. use uniform style; similarly "template fORCDNA synthesis" in line 7 as "template for CDNA synthesis".
We sincerely appreciate your suggestions, we check and corrected whole manuscript for such error.
Please see the attachment.

This manuscript is a resubmission of an earlier submission. The following is a list of the peer review reports and author responses from that submission.
Round 1
Reviewer 1 Report
Dear authors,
I have a great time reveweing the manuscript. There are a few points to consider.
Line 2-3: The title could be reword to give better meaning
Line 5: Numbering for the authors should be in numerical order. Eg. Peihua Gan it should be 1,2 not 1,3.
† symbol is not superscript in the text
Line 14: duplicated the word 'Correspondence'.
Line 18 and others: There are a lot of times authors used 'we', 'our' in the manuscript. Possibly, please avoid using personal pronouns in the manuscript. Please also check for other parts of the manuscript on this matter.
Line 21, 22: there are 3 'were' in a sentence which making the sentence hard to understand.
Line 22: What is ABC. Please specify before using abbreviation. The same comment for Lines 26 and 29 and few other parts in the manuscript.
Line 33: the keywords already used in the title. choose other keyword.
Line 36. In introduction, author can put a Figure/Plate to show what is spiral grain.
Line 40: the sentence start with 'Early'?
I would suggest 'secondary metabolites' incorporated into the Introduction section.
Line 72: 7 thousand can be put in number - 7000.
Line 76: what is 'byat'
Line 77: NR abbreviation was used before mentioning what is NR. This also happens to other abbreviations as well (COG, KEGG, GO, OxoODE, HOTE etc).
Line 160: lactone polykctides - spelling error?
Line 191. Furtherly?
Line 337: No reference for the method?
Line 420 onwards: Some of the references used were old. Please standardize the references according to 'Guidelines to author' for IJMS.
Most of the sentences are easy to understand. Some of the sentences have a few English errors. I would suggest submitting the manuscript to proofread
Reviewer 2 Report
1. Title: It should be more informative to point out the key players and mechanisms for the formation of spiral grains in Pinus yunnanensis.
2. L18: What do you mean by "molecule explains"?
3. L18: Avoid using first-person writing throughout the manuscript.
4. L21-22: It should be rewritten because it's too difficult to follow.
5. L25-30: It should be reorganized to specify the possible regulating mechanisms in more detail, particularly the role of key transcription factors connecting to 2'-deoxyuridine, IAA, and ET-related receptors.
6. L46-54: The description is too weak and needs to be enriched by adding more in-depth insights into the molecular regulation of spiral grain growth of trees from the literature.
7. L206-211: It always needs to design proper statistics for validating the data collected by RT-qPCR particularly based on the required number of replicates and then tested by ANOVA analysis.
8. Figure 6: It should connect to the possible regulating mechanisms in spiral grain formation and subsequent growth.
9. The authors should organize a conclusive graph for presenting the possible regulating mechanisms in spiral grain formation and subsequent growth and for enriching the discussion.
10. More future directions should be added.
Many sentences throughout the manuscript are difficult to follow and need to be rewritten to make them more fluent.